# Systematic Review and Metaanalysis of Worldwide Incidence and Prevalence of Antineutrophil Cytoplasmic Antibody (ANCA) Associated Vasculitis

**DOI:** 10.3390/jcm11092573

**Published:** 2022-05-04

**Authors:** Rocío Redondo-Rodriguez, Natalia Mena-Vázquez, Alba María Cabezas-Lucena, Sara Manrique-Arija, Arkaitz Mucientes, Antonio Fernández-Nebro

**Affiliations:** 1Instituto de Investigación Biomédica de Málaga (IBIMA), UGC de Reumatología, Hospital Regional Universitario de Málaga, 29010 Malaga, Spain; rocioredondo91@hotmail.com (R.R.-R.); albamcabezaslucena@gmail.com (A.M.C.-L.); sarama_82@hotmail.com (S.M.-A.); arkaitzmucientes@gmail.com (A.M.); afnebro@gmail.com (A.F.-N.); 2Departamento de Medicina, Universidad de Málaga, 29016 Malaga, Spain

**Keywords:** ANCA-associated vasculitis, incidence, prevalence

## Abstract

Objective: In this study, we aimed to evaluate the worldwide incidence and prevalence of ANCA-associated vasculitis (AAV). Methods: A systematic search of Medline and Embase was conducted until June 2020 for studies that analyzed the incidence and prevalence of patients aged >16 years diagnosed with AAV in different geographical areas. A meta-analysis was undertaken to estimate the pooled incidence per million person-years and prevalence per million persons in AAV overall and for each subtype of AAV: granulomatosis with polyangiitis (GPA), microscopic polyangiitis (MPA), and eosinophilic granulomatosis with polyangiitis (EGPA). The 95% confidence interval (CI) and I^2^ for heterogeneity were calculated. Results: The meta-analysis included 25 studies that met the inclusion criteria and covered a total of 4547 patients with AAV. Frequency increased over time. The global pooled incidence (95% CI) was 17.2 per million person-years (13.3–21.6) and the global pooled prevalence (95% CI) was 198.0 per million persons (187.0–210.0). The pooled incidence per million person-years for each AAV subtype varied from highest to lowest, as follows: GPA, 9.0; MPA, 5.9; and EGPA, 1.7. The individual pooled prevalence per million persons was, as follows: GPA, 96.8; MPA, 39.2; and EGPA, 15.6. AAV was more predominant in the northern hemisphere. By continent, a higher incidence in America and pooled prevalence of AAV was observed in America and Europe. Conclusion: The pooled incidence and prevalence of AAV seem to be increasing over time and are higher in the case of GPA. AAV was generally more frequent (incidence and prevalence) in the northern hemisphere.

## 1. Introduction

Vasculitis comprises a heterogeneous group of diseases characterized by inflammation and destruction of blood vessel walls, leading to damage in various organs and tissues [1,2]. Antineutrophil cytoplasmic antibody (ANCA)-associated vasculitis (AAV) affects small- and medium-caliber vessels and has a variable clinical course [3]. Its diagnosis is based on clinical factors, serology, and histopathology. The etiology of AAV is unknown, although environmental and genetic factors are involved [4]. The disease is slightly more common in men [5], except for eosinophilic granulomatosis with polyangiitis (EGPA), which is more common in women [6]. AAV includes granulomatosis with polyangiitis (GPA), microscopic polyangiitis (MPA), and EGPA. GPA and MPA are more common than EGPA [7].

Interest in the study of the epidemiology of AAV is increasing, since this will enable us to better understand the factors underlying the disease and enable better health care planning. This meta-analysis is significant for several reasons. First, an accurate and valid estimate of the incidence and prevalence of AAV can support the need for health resources, as well as provide solid evidence for the implementation of general procedures that promote early care for these patients. Second, identifying the differences in incidence and prevalence by geographic area allows us to support that there are different genetic and environmental factors that can influence the pathogenesis of the disease. Therefore, this study provides a basis for future research that could help inform intervention for the evaluation and treatment of patients with this pathology with a high impact on health. While little was known about the frequency of AAV until a relatively short time ago, recent years have seen the reporting of large datasets that have not yet been assessed in a meta-analysis. Furthermore, the incidence and prevalence of AAV in general seem to have increased in recent decades, in part owing to better diagnosis and therapeutic management of these diseases [4].

The objective of this systematic review and meta-analysis was to synthetize current evidence on the incidence and prevalence of AAV and its subtypes throughout the world.

## 2. Materials and Methods

### 2.1. Search Strategy and Selection of Studies

We performed a systematic search in Medline and Embase up to 6 June 2020 using the following MeSH terms and free text: “incidence” or “incidences”, “prevalence” or “prevalences” and “systemic vasculitis” or “ANCA-associated vasculitis” (Appendix A). A secondary manual search of related articles was also performed. This was restricted to English language articles and human studies. The review protocol followed the declaration of Preferred Reporting Items for Systematic Reviews and Meta-Analyses (PRISMA) and was submitted to PROSPERO (being assessed, CRD42020216225). The search was performed by 2 researchers (R-R, R and M-V, N), who independently reviewed article titles and abstracts. Disagreement between reviewers on the inclusion/exclusion of studies was resolved by consensus or with the assistance of a third reviewer (F-N, A).

### 2.2. Inclusion and Exclusion Criteria

The inclusion criteria were as follows: (1) English language articles only; (2) cross-sectional studies, case series, and cohort studies on the incidence and prevalence of AAV in adults (>16 years), with AAV defined according to the 2012 Revised International Chapel Hill Consensus Conference, and the European Medicines Agency Algorithm (EMEA) [8]; and (3) studies that presented a case definition with only data obtained after 1995 due the concept of AAV as group of small vessel vasculitides was not codified until 1994 in Chapel Hill Consensus Conference Document. The exclusion criteria were as follows: (1) editorials, conference abstracts, case reports, or case series with fewer than 30 cases and narrative reviews; (2) insufficient description of methods; (3) lack of data to compute the incidence or prevalence; and (4) duplicate publications.

### 2.3. Outcome Measures

The main outcome measures were the pooled incidence of all cases of AAV, measured as the number of incident cases per million person-years (95% CI) and the pooled prevalence of all AAV measured as the number of prevalent cases per million persons (95% CI). The secondary outcome measures were the pooled incidence and prevalence of each subtype of vasculitis individually and by geographic area.

For the meta-analysis of pooled incidence and pooled prevalence of all types of AAV, we excluded those articles that included in their total count other types of vasculitis, such as polyarteritis nodosa. However, these articles were included for the meta-analysis of each individual type of AAV.

### 2.4. Data Extraction and Measures of Study Quality

The whole text was read in the case of articles whose titles or abstracts met the inclusion criteria. Not fulfilling an eligibility criterion led the study to be excluded. In addition to the main and secondary outcome measures, we extracted information on the authors, year of publication, continent, country, region, age, sex, ethnic group, determination of ANCA, and subtypes. The level of the evidence was assessed using the Scottish Intercollegiate Guidelines Network (SIGN) grading system [9].

### 2.5. Data Synthesis and Analysis

In addition to the calculation of the pooled incidence and the pooled prevalence of the total number of AAV, we calculated the I^2^ statistic as a measure of heterogeneity. In accordance with the Cochrane review rules, we selected the random effects model (DerSimonian and Laird method) in cases of severe heterogeneity with I^2^ > 50%; if this were not the case, then the fixed effects model was chosen (Mantel-Haenszel method) [10]. Sensitivity analyses were conducted by excluding one study at a time.

## 3. Results

### 3.1. Search

The literature search initially retrieved 4577 published studies and 6 articles from a secondary search for the years 1995 to 2020. Of these, 629 were excluded as duplicates, 3901 after reading the title and abstract, and 32 after reading the whole article (Appendix A). Therefore, 25 articles met the selection criteria (Figure 1). Only two included articles used ACR criteria and did not include Chapel Hill criteria have been selected for evaluation independently of GPA or EGPA. All of the studies included in the analysis were observational, most were retrospective, 20 were based on hospital registries, 1 used national surveys, 1 used a national database, 1 combined primary care and hospital databases and data registry from the National Social Security System, and 2 used hospital and primary care registries.

### 3.2. Characteristics of the Studies, Ethnicity, Age, and Sex

Of the 25 studies included, 10 evaluated both incidence and prevalence, 9 only assessed incidence, and 6 only prevalence. Figure 2 shows the types of AAV analyzed in the studies of incidence and/or prevalence, with most evaluating all types of AAV.

A total of 4547 patients were included, with GPA as the most frequent type of AAV. Only 9 of 25 studies reported the ethnicity of the study population, which was Caucasian in most cases.

AAV more commonly affected men than women, and the mean age was higher than 56 years. The ratio of men to women in GPA was 152%, with a mean (SD) age of 57 (5.74) years. The ratio was 164% in MPA, with a mean (SD) age of 65 (5.89) years, and 145% in EGPA, with a mean (SD) age of 52 (7.86) years.

The percentage of ANCA-positive patients was 95% for GPA; these were positive for c-ANCA/anti-PR3 (proteinase 3) in 80%. The percentage of ANCA-positive patients in MPA was 83%, with a predominance of p-ANCA/anti-MPO (myeloperoxidase) (64%). However, only 57% of patients with EGPA were ANCA-positive, with anti-MPO predominating (58%).

### 3.3. Incidence

Table 1 summarizes incidence data for all types of AAV and for each subtype separately throughout the world.

Incidence was analyzed in 19 studies. While eight studies analyzed all types of AAV, one out of the eight also included cases of polyarteritis nodosa. Therefore, only seven studies analyzing only AAV were eventually included.

The pooled incidence of all AAV was 17.2 per million person-years (95% CI, 13.3–21.6) (Figure 3). However, the incidence in the various studies ranged from 8.1 per million (95% CI, 1.0–15.2) [11] to 33.0 per million person-years (95% CI, 24.0–41.0) [12]. The individual pooled incidence for each AAV subtype went from higher to lower in GPA, MPA, and EGPA (Table 2) (See Appendix A).

#### 3.3.1. Geographic Distribution of Incidence

The highest incidence recorded was in Minnesota and the lowest in Turkey. Table 3 shows the pooled incidence of each type of vasculitis by geographic area. Worldwide, the incidence of GPA and MPA was greater in the northern hemisphere than in the southern hemisphere. The incidence of EGPA in the southern hemisphere could not be analyzed since only one study was identified. Analysis by continent revealed that the grouped incidence for GPA was greater in America and that MPA predominated in America and Asia. Rates were similar for EGPA in Europe and Asia, although this could not be calculated for America owing to the lack of studies. The cumulative incidence could not be evaluated in Oceania since only one study was available. No studies from Africa were identified.

#### 3.3.2. Temporal Trends in Incidence

In most studies, we observed an increase in the incidence of AAV over time. Studies from Germany [16], Norway [18], Turkey [11], Taiwan [28], and Minnesota [12] revealed an increasing trend in incidence from the second half of the study period onward. However, the trend remained stable in studies from Oceania [13].

### 3.4. Prevalence

Table 4 summarizes prevalence data for all types of AAV and for each subtype separately throughout the world.

Prevalence was analyzed in 16 studies. While seven of these studies included all types of AAV, we only included four since three of the seven also included cases of polyarteritis nodosa and were therefore excluded.

The pooled prevalence for all types of AAV was 198.0 per million persons (95% CI, 187.0–210.0) (Figure 4). However, the prevalence in the different studies varied from 44.8 per million persons (95% CI, 23.5–66.1) [30] to 421.0 per million persons (95% CI, 296.0–546.0) [12]. The individual pooled prevalence was highest for GPA, followed by MPA and EGPA (Table 5) (See Appendix A).

#### 3.4.1. Geographic Distribution of Prevalence

The highest prevalence reported was from Minnesota and the lowest from south Spain. Table 6 shows the pooled prevalence for each type of vasculitis by geographic area. By hemisphere, both GPA and MPA were more prevalent in the northern hemisphere than in the southern hemisphere. However, EGPA could not be analyzed in the southern hemisphere, since only one study was identified. By continent, the pooled prevalence for GPA was greater in Europe and MPA in America. Asia could not be evaluated since only one study was identified. The prevalence of EGPA was greater in Europe than in Asia, although it could not be evaluated in Oceania or America owing to the lack of studies. No studies from Africa were identified.

#### 3.4.2. Temporal Trends in Prevalence

Studies from Oceania [13] and Norway [18] evaluated prevalence for the different study periods and found that it increased in the second period.

## 4. Discussion

In this study we estimated the pooled incidence and the pooled prevalence of AAV through a systematic review of the literature and meta-analysis. Our results show a pooled incidence of 17.2 per million person-years (95% CI, 13.3–21.6) and a pooled prevalence of 198.0 per million persons (95% CI, 187.0–210.0) in patients with AAV. Nevertheless, we recorded marked variability in the results between studies. Incidence and prevalence generally seemed to increase over time, probably owing to the increased frequency resulting from better identification of cases and greater survival of patients with AAV [37,38].

In our meta-analysis, both incidence and prevalence were greater for GPA than for MPA and EGPA. This finding was consistent with those of most of the individual studies included, although MPA was more common than GPA and EGPA [22,23,25,29,30,33]. The higher incidence and prevalence recorded for GPA in the present meta-analysis could be influenced by the higher number of studies that specifically addressed GPA compared with those that addressed the frequency of AAV overall.

The highest incidence of GPA was reported by Hissaria et al. in 2008 in South Australia [14]; the highest incidence for MPA was reported by Fujimoto et al. in Japan [27]. The highest incidence of EGPA was reported by M.D Alvise-Berti et al. 2017 [12]. In other words, the incidence of one or another type of vasculitis may depend on the geographic area where the study was carried out and, therefore, on genetic or ethnic differences [39,40], as seen in some of the studies included in our analysis. Similarly, retrospective analyses from Asia other than Japan (although it could not be included in this meta-analysis) also indicate MPA seems to be the most common disease subtype among AAV [41], indicating a clear difference (predominance of GPA in the Western countries and MPA in Asia, respectively) according to geographic regions. Likewise, it has been shown that there are differences regarding the phenotype between Europe and Japan. The patients with MPA in Japan had a higher age at onset, more frequent MPO-ANCA positivity, lower serum creatinine, and more frequent interstitial pneumonitis than those in Europe [42].

In addition to the geographic variability by country, our study highlighted variability between the northern and southern hemispheres and between continents, thus, again, highlighting ethnic and/or genetic differences, in addition to differences in latitude [41], as factors that can contribute to the greater incidence in America and greater prevalence in Europe and America.

In general, both the incidence and the prevalence of AAV vary depending on latitude (northern or southern hemisphere) [41], probably due to environmental factors such as the degree of ultraviolet radiation, which varies with latitude within the specific hemisphere [41]. In fact, Gatenby et al. [43,44] found a correlation between GPA, latitude, and levels of ultraviolet radiation. However, our study cannot explain whether the variability of results depends only on geographic regions, on latitude, ethnic, genetic or environmental factors, or even due to factors inherent to access to health care.

The present study is subject to a series of limitations. First, we were unable to determine the global prevalence or incidence of AAV in some studies since the data collected included polyarteritis nodosa. However, we retrieved individual data on AAV subtypes in order to perform an independent analysis of the incidence and prevalence of these conditions. Thus, we carried out the first systematic review and meta-analysis to date to assess the pooled incidence and prevalence of AAV overall and according to the various subtypes. We also analyzed different geographic areas and found that few studies had been performed on some continents, such as Oceania and Asia, and that none had been performed in Africa. Similar results were found when we performed our analysis by hemisphere, with more studies in the northern hemisphere. Even so, we were able to draw comparisons between the northern and southern hemispheres, as well as differences with other continents. Our study did not include papers on the incidence and prevalence of AAVs in children (<16 years). However, it has been described that the prevalence of systemic vasculitis in children accounts for 2 to 10% of rheumatic diseases, IgA vasculitis and Kawasaki disease are the most common, whereas AAV are rarer [45]. Finally, although in our study all articles included AAV defined according to the 2012 Revised International Chapel Hill Consensus Conference, and the European Medicines Agency Algorithm (EMEA), in two included articles used ACR criteria and did not include Chapel Hill criteria have been selected for evaluation independently of GPA or EGPA.

## 5. Conclusions

In conclusion, our results show that the pooled incidence and pooled prevalence of AAV seem to be increasing over time and are higher for GPA. AAV was generally more frequent (incidence and prevalence) in the northern hemisphere.

## Figures and Tables

**Figure 1 jcm-11-02573-f001:**
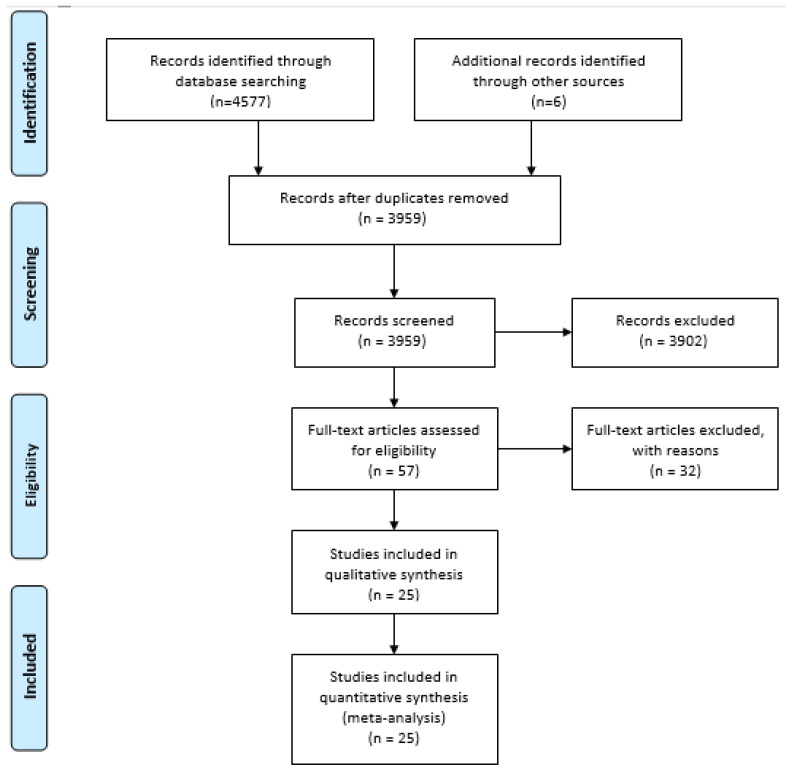
Preferred Reporting Items for Systematic Reviews and Meta-Analyses (PRISMA) flow diagram.

**Figure 2 jcm-11-02573-f002:**
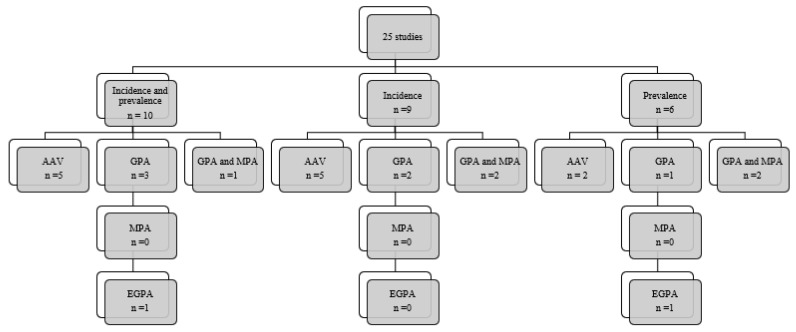
Types of vasculitis analyzed.

**Figure 3 jcm-11-02573-f003:**
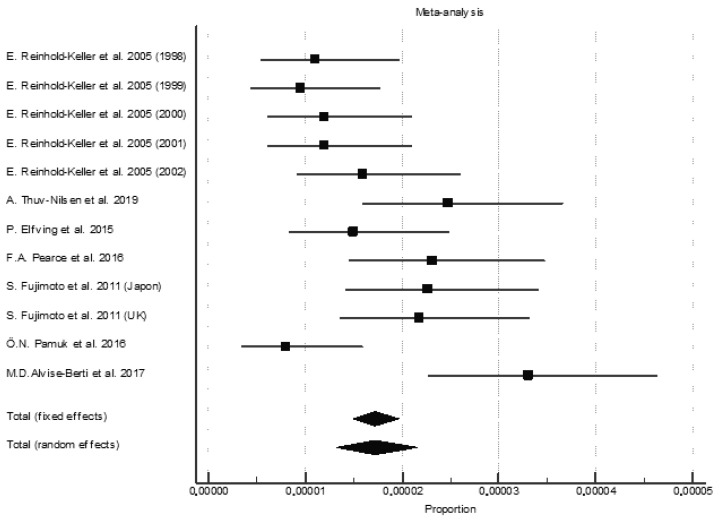
Pooled incidence of total ANCA-associated vasculitis [11,12,16,18,20,25,27].

**Figure 4 jcm-11-02573-f004:**
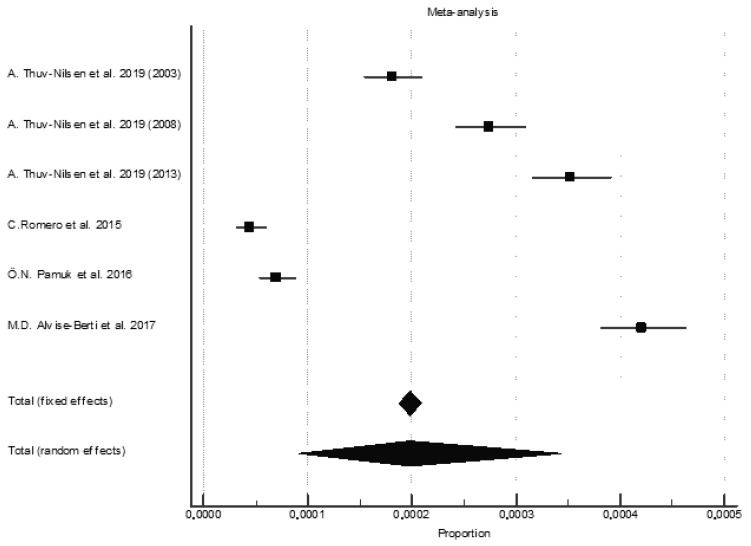
Pooled prevalence of total ANCA-associated vasculitis [11,12,18,30].

**Table 1 jcm-11-02573-t001:** Worldwide incidence of AAV.

Continent	Country	References	Region	Method	Number of Incident Cases	Incidence Per Million Person-Years (95% CI) AAV Overall *	Incidence Per Million Person-Years (95% CI) AAV by Subtype **
Oceania	Australia	A.S. Ormerod et al., 2008 [13]	Canberra and New South Wales	Hospital registry Retrospective	-AAV + PAN: Period 1995–1999: 29 Period 2000–2004: 31	-AAV + PAN: Period 1995–1999: 17.0 Period 2000–2004: 16.2	Period 1995–1999: -GPA 8.8 (4.1–17.1) -MPA 2.3 (0.2–7.2) -EGPA 2.3 (0.6–7.2) Period 2000–2004: -GPA 8.4 (3.5–15.8) -MPA 5.0 (1.6–11.7) -EGPA 2.2 (0.6–7.2)
Oceania	Australia	P. Hissaria et al., 2008 [14]	Southern Australia	Hospital registry Retrospective	Period 2001–2005: -GPA: 84		Period 2001–2005: -GPA: 56 (44.1–68.4)
Europe	Germany	E. Reinhold-Keller et al., 2002 [15]	North and South Germany	Hospital registry Prospective	Period 1998 and 1999: -AAV: 90 -GPA: 61 -MPA: 23 -EGPA: 6		Period 1998: North: -GPA: 8.0 (2–14) -MPA: 3.0 (0–6) -EGPA: 0 South: -GPA: 6.0 (3–9) -MPA: 2.0 (0–4) -EGPA: 1.0 (0–2) Period 1999: North: -GPA: 6.0 (1–11) -MPA: 2.5 (0–6) -EGPA: 1.0 (0–3) South: -GPA: 5.0 (2–8) -MPA: 1.0 (0–2) -EGPA: 1.0 (0–2)
Europe	Germany	E. Reinhold-Keller et al., 2005 [16]	Schleswig-Holstein	Hospital registry Prospective	Period 1998–2002: -AAV: 170	-AAV: 1998: 11 (5–18) 1999: 9.5 (4–16) 2000: 12 (5–19) 2001: 12 (5–19) 2002: 16 (8–24)	-GPA: 1998: 8 (2–14) 1999: 6 (1–11) 2000: 8 (2–13) 2001: 9 (3–15) 2002: 12 (5–19) -MPA: 1998: 3 (0–6) 1999: 2.5 (0–6) 2000: 3 (0–6) 2001: 2 (0–4) 2002: 3 (0–6) -EGPA: 1998: 0 (0) 1999: 1 (0–3) 2000: 1.5 (0–3) 2001: 1 (0–3) 2002: 2 (0–4)
Europe	France	J. Vinit et al., 2009 [17]	Burgundy	Hospital registry Retrospective	Period 1998–2008: -EGPA: 31		Period 1998–2008: -EGPA: 1.2
Europe	Norway	A. Thuv-Nilsen et al., 2019 [18]	Northern Norway (Nordland, Troms, Finnmark)	Hospital registry Retrospective	Period 1999–2013: -AAV: 140	Period 1999–2013 AAV: 24.7 (20.8–29.2)	Period 1999–2013: -GPA 15.6 (12.5–19.2) -MPA 6.5 (4.6–9.0) -EGPA 2.7 (1.5–4.5)
Europe	Poland	K. Kanecki et al., 2018 [19]	Warsaw, Lublin	Hospital registry Retrospective	Period 2011–2015: -GPA: 1491		Period 2011–2015: -GPA: 7.7 (4.1–11.4)
Europe	Finland	P. Elfving et al., 2015 [20]	Northern Savo area	Hospital registry General practice registry Prospective	Period 2010: -AAV: 3	Period 2010: -AAV: 15 (0.3–4.3)	
Europe	Italy	M. Catanoso et al., 2014 [21]	Northern Italy (Reggio Emilia area)	Hospital registry Retrospective	Period 1995–2009: -GPA: 18		Period 1995–2009: -GPA: 2.8 (1.45–4.1)
Europe	Greece	S.H. Panagiotakis et al., 2009 [22]	Crete (southern Greece)	Hospital registry General practice registry Retrospective	Period 1995–2003: -GPA: 21 -MPA: 26		Period 1995–2003: -GPA: 6.6 (3.7–9.6) -MPA: 10.2 (5.8–14.6)
Europe	Sweden	J.M. Aladdin et al., 2009 [23]	Southern Sweden	Hospital registry Prospective	Period 1997–2006: -AAV: 134	Period 1997–2006: -AAV + PAN: 21.8 (18.2–25.4)	Period 1997–2006: -GPA: 9.8 (7.4–12.2) -MPA:10.1 (7.7–12.6) -EGPA: 0.9 (0–1.7)
Europe	United Kingdom	F.A. Pearce et al., 2017 [24]	Nottingham	CPRD (Clinical Practice Research Datalink) HES (Hospital Episode Statistics) Retrospective	Period 1997–2013: -GPA: 462		Period 1997–2013: -GPA: 11.8 (10.7–12.9)
Europe	United Kingdom	F. A. Pearce et al., 2016 [25]	Urban area of Nottingham, Derby	Hospital registry Retrospective	Period 2007–2013: -AAV: 107	Period 2007–2013: -AAV: 23.1 (18.9–27.9)	Period 2007–2013: -GPA 8.2 (5.8–11.3) -MPA 13.4 (10.3–17.2) -EGPA 1.5 (0.6–3.1)
Europe	United Kingdom	R.A. Watts et al., 2012 [26]	Norwich	Hospital registry Retrospective.	Period 1988–2010: -GPA + MPA: 17.2 (14.7–20.0):169	Period 1988–2010: -GPA + MPA: 17.2 (14.7–20.0)	Period 1998–2010: -GPA: 11.3 (9.1–13.4) -MPA: 5.9 (4.4–7.5)
Asia/Europe	Japan/United Kingdom	S. Fujimoto et al., 2011 [27]	Miyazaki (Japan) Norwich (UK)	Hospital registry Prospective	Period 2005–2009: -AAV: Japan: 86 UK: 50	Period 2005–2009: -AAV: Japan: 22.6 (19.1–26.2) UK: 21.8 (12.6–30.9)	Period 2005–2009 Japan: -GPA: 2.1 (0.6–3.7) -MPA: 18.2 (14.3–22.0) -EGPA: 2.4 (0.3–4.4) UK: -GPA: 14.3 (5.8–23.0) -MPA: 6.5 (1.9–11.2) -EGPA: 0.9 (0–1.9)
Asia	Turkey	Ö.N. Pamuk et al., 2016 [11]	Thrace	Hospital registry Retrospective	Period 2004–2014: -AAV:50	Period 2004–2014: -AAV: 8.1 (1–15.2)	Period 2004–2014: -GPA: 4.8 (0–10.3) -MPA: 2.4 (0–6.3) -EGPA: 0.8 (0–4)
Asia	Taiwan	W. Chien-Sheng et al., 2014 [28]	New Taipei city	Hospital registry Retrospective	Period 1997–2008: -GPA: 96		Period 1997–2008: -GPA: 0.37 (0.30–0.45)
America	USA	M.D. Alvise-Berti et al., 2017 [12]	Minnesota (Olmsted County)	Hospital registry Retrospective	Period 1996–2015: -AAV: 58	Period 1996–2015: -AAV: 33 (24–41)	Period 1996–2015: -GPA: 13 (8–18) -MPA: 16 (10–22) -EGPA: 4 (1–6)
America	Argentina	F. S. Pierini et al., 2019 [29]	Buenos Aires	Hospital registry Retrospective	Period 2000–2015: -GPA + MPA: 47 -GPA: 19 -MPA: 28		Period 2000–2015: -GPA: 9 (5–13) -MPA:14 (9–19)

* Incidence per million person-years for ANCA-associated vasculitis overall. ** Incidence per million person-years for each subtype of ANCA-associated vasculitis. Abbreviations: AAV, ANCA-associated vasculitis; GPA, granulomatosis with polyangiitis; MPA, microscopic polyangiitis; EGPA, eosinophilic granulomatosis with polyangiitis; PAN, polyarteritis nodosa.

**Table 2 jcm-11-02573-t002:** Pooled incidence of ANCA-associated vasculitis (number of studies included, 95% CI, and I^2^ [heterogeneity]).

Disease	No. of Studies **	Incidence Per Million/Person-Years (95% CI)	I^2^ (%)
AAV overall *	7	17.2 (13.3–21.6)	67.8
GPA	15	9.0 (7.8–10.3)	39.8
MPA	12	5.9 (4.9–7.0)	71.7
EGPA	10	1.7 (1.2–2.4)	0

* Includes incidence of AAV overall. ** Number of studies included. Abbreviations: AAV, ANCA-associated vasculitis; GPA, granulomatosis with polyangiitis; MPA, microscopic polyangiitis; EGPA, eosinophilic granulomatosis with polyangiitis.

**Table 3 jcm-11-02573-t003:** Pooled incidence of ANCA-associated vasculitis by geographic region (number of studies included, 95% CI, and I^2^ [heterogeneity]).

Hemisphere	Disease	No.* of Studies	Incidence Per Million Person-Years (95% CI)	I^2^ (%)
North	GPA	12	9.5 (8.1–11.0)	0
	MPA	9	7.9 (6.4–9.7)	68.7
	EGPA	9	1.6 (1.0–2.3)	0
South	GPA	2	9.0 (6.1–12.6)	0
	MPA	2	6.8 (1.9–14.8)	79.1
	EGPA	1	-	-
Continent	Disease	No.* of studies	Incidence per million person-years (95% CI)	I^2^ (%)
Oceania	-	1	-	-
Europe	GPA	10	8.5 (7.2–9.9)	16.0
	MPA	7	4.7 (3.1–6.6)	59.8
	EGPA	6	1.7 (1.0–2.7)	0
Asia	GPA	2	3.7 (1.6–7.4)	34
	MPA	2	9.1 (0.1–29.7)	93.4
	EGPA	2	1.9 (0.5–4.3)	0
America	GPA	2	11.4 (7.2–17.1)	0
	MPA	2	15.5 (10.5–22.0)	0
	EGPA	1	-	-

* Number of studies included. Abbreviations: GPA, granulomatosis with polyangiitis; MPA, microscopic polyangiitis; EGPA, eosinophilic granulomatosis with polyangiitis.

**Table 4 jcm-11-02573-t004:** Worldwide prevalence of AAV.

Continent	Country	References	Region	Method	Number of Prevalent Cases	Prevalence Per Million Persons (95% CI) AAV Overall *	Prevalence Per Million Persons (95% CI) AAV by Subtype **
Oceania	Australia	A.S. Ormerod et al., 2008 [13]	Canberra and New South Wales	Hospital registry Retrospective	-AAV + PAN: Period 1995–1999: 41 Period 2000–2004: 67	-AAV + PAN: Period 1995–1999: 114.0 (94–140) Period 2000–2004: 184.4 (158.4–212.6)	Period 1995–1999: -GPA: 64.3 (49.3–81.7) -MPA: 17.5 (10.7–28.4) -EGPA: 11.7 (6.2–29.6) Period 2000–2004: -GPA: 95.0 (76.9–116.1) -MPA: 39.1 (27.7–53.3) -EGPA: 22.3 (13.4–33.3)
Oceania	New Zealand	A. Gibson et al., 2006 [31]	Canterbury	Hospital registry Retrospective	Period 1999–2003: -GPA: 73 -MPA: 28		Period 1999–2003: -GPA: 152 (117–186) -MPA: 58 (37–80) Period 31 December 2003: -GPA: 112 (82–142) -MPA: 37 (20–55)
Europe	Norway	A. Thuv-Nilsen et al., 2019 [18]	North Norway (Nordland, Troms, Finnmark)	Hospital registry Retrospective	Period 1999–2013: -AAV: 140	-AAV: 2003: 181 (141–230) 2008: 274 (223–332) 2013: 352 (296–416)	Periods: 31 December 2003: -GPA: 154 (117–210) -MPA: 8.1 (1.7–23.7) -EGPA: 18.9 (7.6–38.9) 31 December 2008: -GPA: 226 (180–279) -MPA: 29.2 (14.6–52.2) -EGPA: 18.6 (7.5–38.3) 31 December 2013: -GPA: 261 (213–316) -MPA: 58.2 (36.9–87.3) -EGPA: 32.9 (17.5–56.3)
Europe	Denmark	W.W. Eaton et al., 2007 [32]	Denmark	Hospital registry Retrospective	Period 2001: -GPA: 568		Period 2001: -GPA: 100
Europe	Italy	M. Catanoso et al., 2014 [21]	North Italy(Reggio Emilia area)	Hospital registry Retrospective	Period 1995–2009: -GPA: 18		Period 31 December 2009: -GPA: 40.3 (24.7–62.5)
Europe	France	A. Mahr et al., 2004 [33]	Northeast Paris	Hospital registry NHIS (National Health Insurance System) General practice registry Retrospective	Period 2000: -AAV: 45 -GPA: 21 -MPA: 16 -EGPA: 8	Period 2000: -AAV + PAN: 90.3 (74–106)	Period 2000: -GPA: 23.7 (16–31) -MPA: 25.1 (16–34) -EGPA: 10.7 (5–17)
Europe	France	J. Vinit et al., 2009 [17]	Burgundy	Hospital registry Retrospective	Period 1998–2008: -EGPA: 31		Period 1998–2008: -EGPA: 11.3
Europe	Poland	K. Kanecki et al., 2018 [19]	Warsaw, Lublin	Hospital registry Retrospective	Period 2011–2015: -GPA: 6995		Period 31 December 2015: -GPA: 36
Europe	Spain	C. Romero et al., 2015 [30]	Costa del Sol	Hospital registry Retrospective		Period 2010: -AAV: 44.8 (23.5–66.1)	Period 2010: -GPA: 15.8 (3.1–28.4) -MPA: 23.8 (8.2–39.2) -EGPA: 5.3 (0–12.5)
Europe	Sweden	J.M. Aladdin et al., 2007 [34]	South Sweden	Hospital registry General practice registry Cross-sectional	Period 1 January 2003: -AAV: 77 -GPA: 46 -MPA:27 -EGPA: 4	Period 1 January 2003: AAV + PAN: 299 (236–362)	Period 1 January 2003: -GPA: 160 (114–206) -MPA: 94 (58–129) -EGPA: 14 (0.3–27)
Europe	United Kingdom	F.A. Pearce et al., 2017 [24]	Nottingham	CPRD (Clinical Practice Research Datalink) HES (Hospital Episode Statistics) Retrospective	Period 2013: -GPA: 359		Period 2013: -GPA: 134.9 (121.3–149.6)
Asia	Japan	K. Sada et al., 2013 [35]	Japan	Nationwide survey Prospective	Period 2008: -EGPA:1866		Period 2008: -EGPA: 17.8
Asia	Turkey	Ö.N. Pamuk et al., 2016 [11]	Thrace	Hospital registry Retrospective	Period 2013: -AAV: 50	Period November 2013: -AAV: 69.3 (48.6–90)	Period November 2013: -GPA: 41.9 (25.8–58) -MPA: 19.3 (8.4–30.2) -EGPA: 8.1 (1–15.2)
America	USA	M.D. Alvise-Berti et al., 2017 [12]	Minnesota (Olmsted County)	Hospital registry Retrospective	Period 2015: -AAV: 44	Period 1 January 2015: -AAV: 421 (296–546)	Period 2015: -GPA: 218 (129–308) -MPA:184 (101–267) -EGPA: 18 (00–44)
America	USA	A.S. Zeft et al., 2010 [36]	West Montana (Missoula, Kalispell)	Hospital registry Retrospective	Period 2006: -GPA: 32 -MPA: 4		Period 2006: -GPA: 91 (99–101) -MPA: 13 (11–14)
America	Argentina	F.S. Pierini et al., 2019 [29]	Buenos Aires	Hospital registry Retrospective	Period 1 January 2015: -GPA +MPA: 17 -GPA: 10 -MPA: 7		Period 1 January 2015: -GPA 74 (28–120) -MPA: 52 (13–90)

* Prevalence per million persons for ANCA-associated vasculitis overall. ** Prevalence per million for each subtype of ANCA-associated vasculitis. Abbreviations: AAV, ANCA-associated vasculitis; GPA, granulomatosis with polyangiitis; MPA, microscopic polyangiitis; EGPA, eosinophilic granulomatosis with polyangiitis; PAN, polyarteritis nodosa.

**Table 5 jcm-11-02573-t005:** Pooled prevalence of ANCA-associated vasculitis (number of studies included, 95% CI, and I^2^ [heterogeneity]).

Disease	No. of Studies **	Prevalence Per Million Persons (95% CI)	I^2^ (%)
AAV overall *	4	198.0 (187.0–210.0)	99.2
GPA	14	96.8 (92.2–102.0)	98.1
MPA	10	39.2 (35.8–42.7)	96.9
EGPA	9	15.6 (13.4–18.0)	70.0

* Includes the prevalence of AAV overall. ** Number of studies included. Abbreviations: AAV, ANCA-associated vasculitis; GPA, granulomatosis with polyangiitis; MPA, microscopic polyangiitis; EGPA, eosinophilic granulomatosis with polyangiitis.

**Table 6 jcm-11-02573-t006:** Pooled prevalence of ANCA-associated vasculitis by geographic region (number of studies included, 95% CI, and I^2^ [heterogeneity]).

Hemisphere	Disease	No. of Studies *	Prevalence Per Million Persons (95% CI)	I^2^ (%)
North	GPA	11	100.0 (95.0–106.0)	98.5
	MPA	7	40.8 (36.7–45.2)	97.8
	EGPA	8	15.3 (12.9–17.9)	71.9
South	GPA	3	85.8 (77.0–95.4)	81.0
	MPA	3	35.7 (30.1–42.1)	84.1
	EGPA	1	-	-
Continent	Disease	No. of studies *	Prevalence per million persons (95% CI)	I^2^ (%)
Oceania	GPA	2	101.0 (90.1–113.0)	94.6
	MPA	2	30.8 (24.9–37.8)	81.0
	EGPA	1	-	
Europe	GPA	7	137.0 (86.8–198.0)	98.1
	MPA	4	35.6 (16.3–61.9)	95.7
	EGPA	5	15.4 (9.9–22.2)	78.2
Asia	GPA	1	-	-
	MPA	1	-	-
	EGPA	2	12.9 (5.2–24.1)	73
America	GPA	3	121.0 (52.4–217.0)	97.7
	MPA	3	66.7 (57.8–76.6)	99.0
	EGPA	1	-	-

* Number of studies included. Abbreviations: GPA, granulomatosis with polyangiitis; MPA, microscopic polyangiitis; EGPA, eosinophilic granulomatosis with polyangiitis.

## Data Availability

Data presented in this study are available on request from the corresponding author.

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
