# Peer review of "Systematic Review and Metaanalysis of Worldwide Incidence and Prevalence of Antineutrophil Cytoplasmic Antibody (ANCA) Associated Vasculitis"

_jcm, 2022, doi:10.3390/jcm11092573_

Round 1

Reviewer 1 Report

Thank you for submitting this manuscript.

This manuscript in my opinion could have provided more valuable information on the subject if the authors decided to include if not all, much more pertinent bibliography available. They only selected 25 studies out of 3959 publications due to their arbitrarily set inclusion and exclusion criteria. 

The authors included the 3 major clinico pathologic variants  of AAV namely MPA, GPA and EGPA but at the same time neglected to further categorize MPO-ANCA and PR-3 ANCA which relate to MPA & EGPA and, GPA respectively, as depicted in a publication of Kakoullis L et al.  Pertinent bibliography is already available through this above named manuscript and an update would not in my opinion disrupt their findings but instead enhance them.

The authors further note that their results are not supported in a balanced way by enough studies from the northern and southern hemisphere respectively; neither are they balanced if one  examines the studies by continent. The authors note that there is "marked variability in results between studies".

The collected data cannot at this stage reliably depict and explain aetiologically the variability of results; hence the results obtained cannot reliably explain whether the variability of results depend (entirely or even in part) on geographic regions, on latitude, ethnic, genetic or environmental factors. Could it be that pertinent lab exams are not performed as often in the southern hemisphere or in certain continents? Further, one cannot accurately corelate the author's findings with ultraviolet radiation as is suggested in their script; clinically ultraviolet radiation would be expected to corelate with eye central cataract or melanoma and then one would have to report on ultraviolet wavelengths in geographical areas and time of exposure. 

Figure 2 has to be re-written in English in some instances in the depiction provided. 

On line 238, the word "were" needs to be corrected to "where".

References 23 and 67 are duplicate?

Between ref 44 and 45 the word "References" must be erased. 

Overall, my opinion is that this script can be re-written and possibly get published as a letter or as a short communication with a view to encourage researchers to provide more focused research on this subject which eventually could lead to a meaningful metanalysis.

Author Response

Comments for the reviewers

We would like to thank the editor for considering our work for publication in “Journal of Clinical Medicine” and the reviewers for their comments, which have helped to improve the quality of our manuscript.

Below, we provide a point-by-point reply to the comments.

Reviewer #1

  1. This manuscript in my opinion could have provided more valuable information on the subject if the authors decided to include if not all, much more pertinent bibliography available. They only selected 25 studies out of 3959 publications due to their arbitrarily set inclusion and exclusion criteria. 

Reply: The concept of AAV as group of small vessel vasculitides comprising GPA, MPA and EGPA was formed during the first part of the 1990ies. It was fist codified in the Chapel Hill Consensus Conference Document published 1994. It took a varying amount of time until these concepts and disease categories were generally embraced by rheumatologists and nephrologists in different countries. This was paralleled by the spread of ANCA testing. With this in mind, epidemiological data on AAV from the first of half of the 1990-ies are of very little value. Therefore, we have only included data obtained after 1995 should be included in the analysis. This has been described in the inclusion and exclusion criteria of the document.

-Pag 2, line 68-78: Inclusion and exclusion criteria: “The inclusion criteria were as follows: (1) English language articles only; (2) cross-sectional studies, case series, and cohort studies on the incidence and prevalence of AAV in adults (>16 years), with AAV defined according to the 2012 Revised International Chapel Hill Consensus Conference, and the European Medicines Agency Algorithm (EMEA) (8); and (3) studies that presented a case definition with only data obtained after 1995 due the concept of AAV as group of small vessel vasculitides was not codified until 1994 in Chapel Hill Consensus Conference Document. The exclusion criteria were as follows: (1) editorials, conference abstracts, case reports, or case series with fewer than 30 cases and narrative reviews; (2) insufficient description of methods; (3) lack of data to compute the incidence or prevalence; and (4) duplicate publications.”

  1. The authors included the 3 major clinico pathologic variants  of AAV namely MPA, GPA and EGPA but at the same time neglected to further categorize MPO-ANCA and PR-3 ANCA which relate to MPA & EGPA and, GPA respectively, as depicted in a publication of Kakoullis L et al.  Pertinent bibliography is already available through this above named manuscript and an update would not in my opinion disrupt their findings but instead enhance them.

Reply: In the publication of Kakoullis L et al., they conducted a systematic review to identify cases of infection-induced anti-myeloperoxidase (MPO) antineutrophil cytoplasmic antibody (ANCA)-associated vasculitis (AAV). We have not included this article in our study because our objective was to know the incidence and prevalence of ANCA vasculitis. Our objective is not to identify cases of infection-induced anti-myeloperoxidase (MPO) antineutrophil cytoplasmic antibody (ANCA)-associated vasculitis (AAV). In addition, as can be seen in the exclusion criteria, this article would be excluded by the following criteria described: (3) lack of data to compute the incidence or prevalence.

-Pag 2, line 68-78: Inclusion and exclusion criteria: “The inclusion criteria were as follows: (1) English language articles only; (2) cross-sectional studies, case series, and cohort studies on the incidence and prevalence of AAV in adults (>16 years), with AAV defined according to the 2012 Revised International Chapel Hill Consensus Conference, and the European Medicines Agency Algorithm (EMEA) (8); and (3) studies that presented a case definition with only data obtained after 1995 due the concept of AAV as group of small vessel vasculitides was not codified until 1994 in Chapel Hill Consensus Conference Document. The exclusion criteria were as follows: (1) editorials, conference abstracts, case reports, or case series with fewer than 30 cases and narrative reviews; (2) insufficient description of methods; (3) lack of data to compute the incidence or prevalence; and (4) duplicate publications.”

  1. The authors further note that their results are not supported in a balanced way by enough studies from the northern and southern hemisphere respectively; neither are they balanced if one  examines the studies by continent. The authors note that there is "marked variability in results between studies".

Reply: As the reviewer says our results are not supported in a balanced way by enough studies from the northern and southern hemisphere respectively. However, the main objective of our study was to assess the pooled incidence and prevalence of AAV in general and according to the different subtypes. Furthermore, this is the first systematic review and meta-analysis to date to assess the pooled incidence and prevalence of AAV. Even so, we were able to draw comparisons between the northern and southern hemispheres, as well as differences with other continents. This limitation has been described in the discussion. Also, their strengths.

Pag. 18, line 260-265: “We also analyzed different geographic areas and found that few studies had been performed on some continents, such as Oceania and Asia, and that none had been performed in Africa. Similar results were found when we performed our analysis by hemisphere, with more studies in the northern hemisphere. Even so, we were able to draw comparisons between the northern and southern hemispheres, as well as differences with other continents.”

  1. The collected data cannot at this stage reliably depict and explain aetiologically the variability of results; hence the results obtained cannot reliably explain whether the variability of results depend (entirely or even in part) on geographic regions, on latitude, ethnic, genetic or environmental factors. Could it be that pertinent lab exams are not performed as often in the southern hemisphere or in certain continents? Further, one cannot accurately corelate the author's findings with ultraviolet radiation as is suggested in their script; clinically ultraviolet radiation would be expected to corelate with eye central cataract or melanoma and then one would have to report on ultraviolet wavelengths in geographical areas and time of exposure

Reply: We have based ourselves on the bibliography described. In the study by Paul A. Gatenby et al. describes and quantifies the association between ambient ultraviolet (UV) radiation levels, and the incidence of the 3 antineutrophil cytoplasmic antibody-associated vasculitides (AAVs). The incidence of AAVs increased with increasing latitude and decreasing ambient UV radiation. However, we understand that our study cannot explain whether the variability of results depends only on geographic regions, on latitude, ethnic, genetic or environmental factors or even due to factors inherent to access to health care. We have added this comment to the discussion.

Pag. 18, line 253-255: “In general, both the incidence and the prevalence of AAV vary depending on latitude (northern or southern hemisphere) (29), probably because of environmental factors such as the degree of ultraviolet radiation, which varies with latitude within the specific hemisphere (29). In fact Gatenby et al. (30,31) found a correlation between GPA, latitude, and levels of ultraviolet radiation. However, our study cannot explain whether the variability of results depends only on geographic regions, on latitude, ethnic, genetic or environmental factors, or even due to factors inherent to access to health care.”

  1. Figure 2 has to be re-written in English in some instances in the depiction provided. 

Reply: We appreciate your comment. We have corrected Figure 2.

Pag. 5: Figure 2. Types of vasculitis analyzed.

  1. On line 238, the word "were" needs to be corrected to "where".

Reply: We appreciate your comment. We have corrected this Word.

Pag. 17; line 238: “In other words, the incidence of one or another type of vasculitis may depend on the geographic area where the study was carried out and, therefore, on genetic or ethnic differences (27,28), “

  1. References 23 and 67 are duplicate?

Reply: We appreciate your comment. We have modified the bibliography that was mistakenly duplicated.

Pag. 39-42. References

  1. Jennette JC. Chapel-Hill Vasculitis 2012. Clin Exp Nephrol. 2013;17(5):603–6.
  2. Gapud EJ, Seo P, Antiochos B. ANCA-Associated Vasculitis Pathogenesis: A Commentary. Curr Rheumatol Rep. 2017;19(4):1–7.
  3. Santiago Rivero D. Enfrentamiento de las vasculitis primarias. Rev Médica Clínica Las Condes [Internet]. 2012;23(4):403–11. Available from: http://dx.doi.org/10.1016/S0716-8640(12)70331-0
  4. Berti A, Dejaco C. Update on the epidemiology, risk factors, and outcomes of systemic vasculitides. Best Pract Res Clin Rheumatol [Internet]. 2018 Apr 1 [cited 2020 Jul 29];32(2):271–94. Available from: http://www.ncbi.nlm.nih.gov/pubmed/30527432
  5. Geetha D, Jefferson J. ANCA-Associated Vasculitis: Core Curriculum 2020. Am J Kidney Dis [Internet]. 2020 Jan;75(1):124–37. Available from: http://www.embase.com/search/results?subaction=viewrecord&from=export&id=L2002390148
  6. Castellano Cuesta JA, González Domínguez J, García Manzanares A. Sindrome De Churg-Strauss. Arq Med. 1993;7(6):350–4.
  7. Valbuena RJJ, Cantillo TJJ, Contreras VKM. Vasculitis sistémicas y riñón: aproximación teórico-práctica para el clínico. 2012;
  8. Khan I, Watts RA. Classification of ANCA-associated vasculitis. Curr Rheumatol Rep. 2013;15(12):13–8.
  9. Miller J. The Scottish Intercollegiate Guidelines Network (SIGN). Br J Diabetes Vasc Dis. 2002;2(1):47–9.
  10. Cumpston M, Li T, Page MJ, Chandler J, Welch VA, Higgins JP, et al. Updated guidance for trusted systematic reviews: a new edition of the Cochrane Handbook for Systematic Reviews of Interventions. Cochrane database Syst Rev. 2019;10:ED000142.
  11. Pamuk ÖN, Dönmez S, Calayır GB, Pamuk GE. The epidemiology of antineutrophil cytoplasmic antibody-associated vasculitis in northwestern Turkey. Clin Rheumatol. 2016 Aug;35(8):2063–71.
  12. Alvise Berti, M.D., Divi Cornec, M.D. Ph.D., Cynthia S. Crowson, M.S. US, M.D., and Eric L. Matteson, M.D. MPH. he epidemiology of ANCA associated vasculitis in Olmsted County, Minnesota (USA): a 20 year population-based study. 2018;69(12):2338–50.
  13. Reinhold Keller E, Herlyn K, Wagner Bastmeyer R, Gross WL. Stable incidence of primary systemic vasculitides over five years: Results from the German vasculitis resister. Arthritis Care Res [Internet]. 2005;53(1):93–9. Available from: http://www.embase.com/search/results?subaction=viewrecord&from=export&id=L40216357
  14. Nilsen AT, Karlsen C, Bakland G, Watts R, Luqmani R, Koldingsnes W. Increasing incidence and prevalence of ANCA-associated vasculitis in Northern Norway. Rheumatol (United Kingdom). 2020;59(9):2316–24.
  15. Wu CS, Hsieh CJ, Peng Y Sen, Chang TH, Wu ZY. Antineutrophil cytoplasmic antibody-associated vasculitis in Taiwan: A hospital-based study with reference to the population-based National Health Insurance database. J Microbiol Immunol Infect. 2015;48(5):477–82.
  16. Ormerod AS, Cook MC. Epidemiology of primary systemic vasculitis in the Australian Capital Territory and south-eastern New South Wales. Intern Med J. 2008 Nov;38(11):816–23.
  17. Romero Gómez C, Aguilar García JA, García de Lucas MD et al. Epidemiological study of primary systemic vasculitides among adults in southern Spain and review of the main epidemiological studies. Clin Exp Rheumatol [Internet]. 2015;33(2):S-11-8. Available from: http://www.embase.com/search/results?subaction=viewrecord&from=export&id=L613657990
  18. Nelveg-kristensen KE, Szpirt W, Carlson N, Mcclure M, Jayne D, Dieperink H, et al. Increasing incidence and improved survival in ANCA-associated vasculitis — a Danish nationwide study. 2020;1–9.
  19. Watts RA, Mahr A, Mohammad AJ, Gatenby P, Basu N, Flores-Suárez LF, et al. Classification, epidemiology and clinical subgrouping of antineutrophil cytoplasmic antibody (ANCA)-associated vasculitis. Nephrol Dial Transplant [Internet]. 2015 Apr;30:i14–22. Available from: http://www.embase.com/search/results?subaction=viewrecord&from=export&id=L603670601
  20. Pearce FA, Lanyon PC, Grainge MJ, Shaunak R, Mahr A, Hubbard RB, et al. Incidence of ANCA-associated vasculitis in a UK mixed ethnicity population. Rheumatol (United Kingdom) [Internet]. 2016 Sep;55(9):1656–63. Available from: http://www.embase.com/search/results?subaction=viewrecord&from=export&id=L613312860
  21. Mahr A, Guillevin L, Poissonnet M, Aymé S. Prevalences of Polyarteritis Nodosa, Microscopic Polyangiitis, Wegener’s Granulomatosis, and Churg-Strauss Syndrome in a French Urban Multiethnic Population in 2000: A Capture-Recapture Estimate. Arthritis Care Res. 2004;51(1):92–9.
  22. Pierini FS, Scolnik M, Scaglioni V, Mollerach F, Soriano ER. Incidence and prevalence of granulomatosis with polyangiitis and microscopic polyangiitis in health management organization in Argentina: a 15-year study. Clin Rheumatol. 2019 Jul;38(7):1935–40.
  23. Panagiotakis SH, Perysinakis GS, Kritikos H, Vassilopoulos D, Vrentzos G, Linardakis M, et al. The epidemiology of primary systemic vasculitides involving small vessels in Crete (southern Greece): a comparison of older versus younger adult patients. Clin Exp Rheumatol. 2009;27(3):409–15.
  24. Mohammad AJ, Jacobsson LTH, Westman KWA, Sturfelt G, Segelmark M. Incidence and survival rates in Wegener’s granulomatosis, microscopic polyangiitis, Churg-Strauss syndrome and polyarteritis nodosa. Rheumatology (Oxford). 2009 Dec;48(12):1560–5.
  25. Hissaria P, Cai FZJ, Ahern M, Smith M, Gillis D. Wegener ’ s granulomatosis : epidemiological and clinical features in a South Australian study. 2008;38:776–80.
  26. Fujimoto S, Watts RA, Kobayashi S, Suzuki K, Jayne DRW, Scott DGI, et al. Comparison of the epidemiology of anti-neutrophil cytoplasmic antibody-associated vasculitis between Japan and the U.K. Rheumatology (Oxford). 2011 Oct;50(10):1916–20.
  27. Tsuchiya N. Genetics of ANCA-associated vasculitis in Japan: a role for HLA-DRB1*09:01 haplotype. Clin Exp Nephrol. 2013 Oct;17(5):628–30.
  28. Naidu GSRSNK, Prasanna D. Is granulomatosis with polyangiitis in Asia different from the West ? 2018;(August):1–5.
  29. Li J, Cui Z, Long J, Huang W, Wang J, Wang H, et al. The frequency of ANCA-associated vasculitis in a national database of hospitalized patients in China. 2018;1–10.
  30.  Furuta S, Chaudhry AN, Hamano Y, Fujimoto S, Nagafuchi H, Makino H, Matsuo S, Ozaki S, Endo T, Muso E, et al. Comparison of phenotype and outcome in microscopic polyangiitis between Europe and Japan. J Rheumatol. 2014 Feb;41(2):325-33. doi: 10.3899/jrheum.130602.
  31. 31. Gatenby PA, Lucas RM, Engelsen OLA, Ponsonby A, Clements M, Gatenby PA, et al. Antineutrophil Cytoplasmic Antibody – Associated Vasculitides : Could Geographic Patterns Be Explained by Ambient Ultraviolet Radiation ? 2009;61(10):1417–24.
  32. 32. Gatenby PA. Anti-neutrophil cytoplasmic antibody-associated systemic vasculitis: Nature or nurture? Vol. 42, Internal Medicine Journal. 2012. p. 351–9.
  33. Iudici M, Quartier P, Terrier B, Mouthon L, Guillevin L, Puéchal X. Childhood-onset granulomatosis with polyangiitis and microscopic polyangiitis: systematic review and meta-analysis. Orphanet J Rare Dis. 2016 Oct 22;11(1):141. doi: 10.1186/s13023-016-0523-y.
  34. 34. Reinhold Keller E, Herlyn K, Wagner Bastmeyer R, Gutfleisch J ,Peter HH, Raspe HH,Gross WL. No difference in the incidences of vasculitides between north and south Germany: First results of the German vasculitis register. Rheumatology [Internet]. 2002 May;41(5):540–9. Available from: http://www.embase.com/search/results?subaction=viewrecord&from=export&id=L34575305
  35. 35. Vinit J, Muller G, Bielefeld P, Pfitzenmeyer P, Bonniaud P, Lorcerie B, et al. Churg-Strauss syndrome: retrospective study in Burgundian population in France in past 10 years. Rheumatol Int. 2011 May;31(5):587–93.
  36. 36. Kanecki K, Nitsch-Osuch A, Gorynski P, Tarka P, Bogdan M, Tyszko P. Epidemiology of Granulomatosis with Polyangiitis in Poland, 2011-2015. Adv Exp Med Biol. 2018;1116:131–8.
  37. 37. Elfving P, Marjoniemi O, Niinisalo H, Kononoff A, Arstila L, Savolainen E, et al. Estimating the incidence of connective tissue diseases and vasculitides in a defined population in Northern Savo area in 2010. Rheumatol Int. 2016 Jul;36(7):917–24.
  38. 38. Catanoso M, Macchioni P, Boiardi L, Manenti L, Tumiati B, Cavazza A, et al. Epidemiology of granulomatosis with polyangiitis (Wegener’s granulomatosis) in Northern Italy: A 15-year population-based study. Semin Arthritis Rheum [Internet]. 2014;44(2):202–7. Available from: http://dx.doi.org/10.1016/j.semarthrit.2014.05.005
  39. 39. Pearce FA, Grainge MJ, Lanyon PC, Watts RA, Hubbard RB. The incidence, prevalence and mortality of granulomatosis with polyangiitis in the UK Clinical Practice Research Datalink. Rheumatol (United Kingdom). 2017;56(4):589–96.
  40. 40. Watts RA, Mooney J, Skinner J, Scott DGI, Macgregor, A J. The contrasting epidemiology of granulomatosis with polyangiitis (Wegener’s) and microscopic polyangiitis. Rheumatology [Internet]. 2012 May;51(5):926–31. Available from: http://www.embase.com/search/results?subaction=viewrecord&from=export&id=L364687781
  41. 41. Gibson A, Stamp LK, Chapman PT, Donnell JLO. The epidemiology of Wegener ’ s granulomatosis and microscopic polyangiitis in a Southern Hemisphere region. 2006;(December 2005):624–8.
  42. 42. Eaton WW, Rose NR, Kalaydjian A, Pedersen MG, Mortensen PB. Epidemiology of autoimmune diseases in Denmark. J Autoimmun. 2007;29(1):1–9.
  43. 43. Mohammad AJ, Jacobsson LTH, Mahr AD, Sturfelt G, Segelmark M. Prevalence of Wegener’s granulomatosis, microscopic polyangiitis, polyarteritis nodosa and Churg-Strauss syndrome within a defined population in southern Sweden. Rheumatology (Oxford). 2007 Aug;46(8):1329–37.
  44. 44. Sada KE, Amano K, Uehara R, Yamamura M, Arimura Y, Nakamura Y, et al. A nationwide survey on the epidemiology and clinical features of eosinophilic granulomatosis with polyangiitis (Churg-Strauss) in Japan. Mod Rheumatol. 2014;24(4):640–4.
  45. 45. Zeft AS, Schlesinger MH, Keenan H, Larson R, Emery H, Weiss NS. e ci al us er om m. 2015;(March 2010).

  1. Between ref 44 and 45 the word "References" must be erased. 

Reply: We appreciate your comment. We have modified the bibliography that was mistakenly duplicated.

Pag. 39-42. References. References have been modified. It is shown in the previous section.

  1. Overall, my opinion is that this script can be re-written and possibly get published as a letter or as a short communication with a view to encourage researchers to provide more focused research on this subject which eventually could lead to a meaningful metanalysis.

Reply: In our opinion this is the first systematic review and meta-analysis to date to assess the pooled incidence and prevalence of AAV. We have followed a rigorous methodology based on performed a systematic search and meta-analysis of pooled incidence and pooled prevalence. We have given the search data, clinical data, and data summaries. We have based ourselves on inclusion and exclusion criteria to obtain quality data. We have responded to our main objective to assess the pooled incidence and prevalence of AAV in general and according to the different subtypes. For this reason, we believe that the article should be published as an original article.

Reviewer #2

The authors conducted a systematic review to evaluate the worldwide incidence and prevalence of ANCA-associated vasculitis.

The manuscript is well written and successfully synthetizes the current evidence on the epidemiology of of AAV and its subtypes.

Therefore, I have only minor concerns as follows.

Minor concerns:

  1. pg 1, line 36: Why "interstitial Vasculitis"?

Reply: We appreciate your comment. This word has been an error and we have modified it.

Pag.1, line 36-37: “Vasculitis comprises a heterogeneous group of diseases characterized by inflammation and destruction of blood vessel walls, leading to damage in various organs and tissues (1,2).”

  1. pg4, line 132: "with MPA predominating" - I think you mean MPO-ANCA instead of MPA?

Reply: We appreciate your comment. This word has been an error and we have modified it.

Pag. 4, line 132: “The percentage of ANCA-positive patients in MPA was 83 %, with a predominance of p-ANCA/anti-MPO (myeloperoxidase) (64 %). However, only 57 % of patients with EGPA were ANCA-positive, with anti-MPO predominating (58 %).”

  1. Shouldn't figure 2 be labeled in English?

Reply: Figure 2. Types of vasculitis analyzed.

  1. pg 17, line 235: There are differences regarding the phenotype between Europe and Japan (see Futura J Rheumatology 2014), which should be mentioned and further discussed in the context of the higher MPA incidence in Japan (which is already done partly in lines 236-242).

Reply: We appreciate your comment. We have added this comment and its reference.

-Pag. 17, 243-246: “Likewise, it has been shown that there are differences regarding the phenotype between Europe and Japan. The patients with MPA in Japan had a higher age at onset, more frequent MPO-ANCA positivity, lower serum creatinine, and more frequent interstitial pneumonitis than those in Europe (30).”

-References

  1. Furuta S, Chaudhry AN, Hamano Y, Fujimoto S, Nagafuchi H, Makino H, Matsuo S, Ozaki S, Endo T, Muso E, et al. Comparison of phenotype and outcome in microscopic polyangiitis between Europe and Japan. J Rheumatol. 2014 Feb;41(2):325-33. doi: 10.3899/jrheum.130602.

  1. Maybe you should briefly discuss the incidence/prevalence in children.

Reply: In our study, one of the inclusion criteria was to study the incidence and prevalence of AAV in adults (>16 years). However, in the discussion we have briefly added the incidence/prevalence in children.

-Pag 2, line 68-78: Inclusion and exclusion criteria: “The inclusion criteria were as follows: (1) English language articles only; (2) cross-sectional studies, case series, and cohort studies on the incidence and prevalence of AAV in adults (>16 years), with AAV defined according to the 2012 Revised International Chapel Hill Consensus Conference, and the European Medicines Agency Algorithm (EMEA) (8);”

-Pag, line: “Our study did not include papers on the incidence and prevalence of AAVs in children (<16 years). However, it has been described that the prevalence of systemic vasculitis in children accounts for 2 to 10% of rheumatic diseases, IgA vasculitis and Kawasaki disease are the most common, whereas AAV are rarer (33).”

Thank you in advance for your time and consideration.

Reviewer 2 Report

The authors conducted a systematic review to evaluate the worldwide incidence and prevalence of ANCA-associated vasculitis.

The manuscript is well written and successfully synthetizes the current evidence on the epidemiology of of AAV and its subtypes.

Therefore, I have only minor concerns as follows.

Minor concerns:

pg 1, line 36: Why "interstitial Vasculitis"?

pg4, line 132: "with MPA predominating" - I think you mean MPO-ANCA instead of MPA?

Shouldn't figure 2 be labeled in English?

pg 17, line 235: There are differences regarding the phenotype between Europe and Japan (see Futura J Rheumatology 2014), which should be mentioned and further discussed in the context of the higher MPA incidence in Japan (which is already done partly in lines 236-242).

Maybe you should briefly discuss the incidence/prevalence in children.

Author Response

Reviewer #2

The authors conducted a systematic review to evaluate the worldwide incidence and prevalence of ANCA-associated vasculitis.

The manuscript is well written and successfully synthetizes the current evidence on the epidemiology of of AAV and its subtypes.

Therefore, I have only minor concerns as follows.

Minor concerns:

  1. pg 1, line 36: Why "interstitial Vasculitis"?

Reply: We appreciate your comment. This word has been an error and we have modified it.

Pag.1, line 36-37: “Vasculitis comprises a heterogeneous group of diseases characterized by inflammation and destruction of blood vessel walls, leading to damage in various organs and tissues (1,2).”

  1. pg4, line 132: "with MPA predominating" - I think you mean MPO-ANCA instead of MPA?

Reply: We appreciate your comment. This word has been an error and we have modified it.

Pag. 4, line 132: “The percentage of ANCA-positive patients in MPA was 83 %, with a predominance of p-ANCA/anti-MPO (myeloperoxidase) (64 %). However, only 57 % of patients with EGPA were ANCA-positive, with anti-MPO predominating (58 %).”

  1. Shouldn't figure 2 be labeled in English?

Reply: Figure 2. Types of vasculitis analyzed.

  1. pg 17, line 235: There are differences regarding the phenotype between Europe and Japan (see Futura J Rheumatology 2014), which should be mentioned and further discussed in the context of the higher MPA incidence in Japan (which is already done partly in lines 236-242).

Reply: We appreciate your comment. We have added this comment and its reference.

-Pag. 17, 243-246: “Likewise, it has been shown that there are differences regarding the phenotype between Europe and Japan. The patients with MPA in Japan had a higher age at onset, more frequent MPO-ANCA positivity, lower serum creatinine, and more frequent interstitial pneumonitis than those in Europe (30).”

-References

  1. Furuta S, Chaudhry AN, Hamano Y, Fujimoto S, Nagafuchi H, Makino H, Matsuo S, Ozaki S, Endo T, Muso E, et al. Comparison of phenotype and outcome in microscopic polyangiitis between Europe and Japan. J Rheumatol. 2014 Feb;41(2):325-33. doi: 10.3899/jrheum.130602.

  1. Maybe you should briefly discuss the incidence/prevalence in children.

Reply: In our study, one of the inclusion criteria was to study the incidence and prevalence of AAV in adults (>16 years). However, in the discussion we have briefly added the incidence/prevalence in children.

-Pag 2, line 68-78: Inclusion and exclusion criteria: “The inclusion criteria were as follows: (1) English language articles only; (2) cross-sectional studies, case series, and cohort studies on the incidence and prevalence of AAV in adults (>16 years), with AAV defined according to the 2012 Revised International Chapel Hill Consensus Conference, and the European Medicines Agency Algorithm (EMEA) (8);”

-Pag, line: “Our study did not include papers on the incidence and prevalence of AAVs in children (<16 years). However, it has been described that the prevalence of systemic vasculitis in children accounts for 2 to 10% of rheumatic diseases, IgA vasculitis and Kawasaki disease are the most common, whereas AAV are rarer (33).”

Thank you in advance for your time and consideration.

Round 2

Reviewer 1 Report

No added comments

Author Response

Comments for the editor

We would like to thank the editor for considering our work for publication in “Journal of Clinical Medicine”.

Below, we provide a point-by-point reply to the comments.

Academic Editor Notes

  1. In their introduction the authors mention that further knowledge about the epidemiology of AAV could help to furtther understand factors underlying the disease and to improve health care planning of patients with AAV.
    Please further comment how the reported findings in the manuscript might improve understanding of AAV and health care or therapy.  

Reply: Following the recommendation, we have included how the reported findings in the manuscript might improve understanding of AAV and health care or therapy.

-Pag 2, line 48-55: “This meta-analysis is significant for several reasons. First, an accurate and valid estimate of the incidence and prevalence of AAV can support the need for health resources, as well as provide solid evidence for the implementation of general procedures that promote early care for these patients. Second, identifying the differences in incidence and prevalence by geographic area allows us to support that there are different genetic and environmental factors that can influence the pathogenesis of the disease. Therefore, this study provides a basis for future research that could help inform intervention for the evaluation and treatment of patients with this pathology with a high impact on health.”

Thank you in advance for your time and consideration.
